# Functions of Dendritic Cells and Its Association with Intestinal Diseases

**DOI:** 10.3390/cells10030583

**Published:** 2021-03-06

**Authors:** Ze-Jun Yang, Bo-Ya Wang, Tian-Tian Wang, Fei-Fei Wang, Yue-Xin Guo, Rong-Xuan Hua, Hong-Wei Shang, Xin Lu, Jing-Dong Xu

**Affiliations:** 1Clinical Medicine of “5 + 3” Program, Capital Medical University, Beijing 100069, China; balder@ccmu.edu.cn (Z.-J.Y.); wffbest1412@163.com (F.-F.W.); andrewhdd@126.com (R.-X.H.); 2Undergraduate Student of 2018 Eight Years Program of Clinical Medicine, Peking University Health Science Center, Beijing 100081, China; 1810301208@pku.edu.cn; 3Department of Physiology and Pathophysiology, School of Basic Medical Sciences, Capital Medical University, Beijing 100069, China; wtt9400@163.com; 4Oral Medicine of “5 + 3” Program, Capital Medical University, Beijing 100069, China; gyxin2014@163.com; 5Morphological Experiment Center, School of Basic Medical Sciences, Capital Medical University, Beijing 100069, China; hongwei@ccmu.edu.cn (H.-W.S.); luxin@ccmu.edu.cn (X.L.)

**Keywords:** dendritic cell, immune tolerance, IBD, intestinal tumor, intestinal flora

## Abstract

Dendritic cells (DCs), including conventional DCs (cDCs) and plasmacytoid DCs (pDCs), serve as the sentinel cells of the immune system and are responsible for presenting antigen information. Moreover, the role of DCs derived from monocytes (moDCs) in the development of inflammation has been emphasized. Several studies have shown that the function of DCs can be influenced by gut microbes including gut bacteria and viruses. Abnormal changes/reactions in intestinal DCs are potentially associated with diseases such as inflammatory bowel disease (IBD) and intestinal tumors, allowing DCs to be a new target for the treatment of these diseases. In this review, we summarized the physiological functions of DCs in the intestinal micro-environment, their regulatory relationship with intestinal microorganisms and their regulatory mechanism in intestinal diseases.

## 1. Introduction

In 1973, Steinman and Cohn described a population of cells in the spleen of mice that exhibited a different cellular appearance and behavior from that of monocytes and Mφs, which were hence named dendritic cells (DCs) [1]. DCs were observed to be highly capable of initiating and regulating immune responses with a high level expression of major histocompatibility complex II (MCH-II) and integrin X (complement component 3 receptor 4 subunit) [2,3]. As professional antigen-presenting cells (APC), DCs interact with other innate and adaptive immune cells to ensure the specificity of the adaptive immune response. DCs can largely recognize pathogen-associated molecular patterns (PAMPs) through many receptors, like toll-like receptors (TLRs), which bind to a large number of molecules produced by bacteria, viruses, and fungi. At the same time, circulating DCs have been proven to exhibit intestinal homing characteristics and were found to be regulated by different gliadin-derived peptides in celiac patients [4,5]. During the process of embryonic development and postnatally, DC progenitors migrate into non-lymphoid organs so as to differentiate into immature DCs, which gradually develop a dense network of sentinel cells from the body to the viscera [6].

Many studies have demonstrated that several common diseases are associated with changes in DC distribution or functions [7,8,9,10,11,12]. DCs can be regulated to restore T-cell tolerance and modulate autoantibody production, and functional heterogeneity exists within each DC subset [7]. For example, it was shown that the function of DCs lacking TIR-domain-containing adapter-inducing interferon-β (TRIF) is altered and T cell activation is reduced, which facilitates the progression of diabetes induced by diabetogenic T cells [8]. DCs were also proven to have an influence on the process of diseases like transmissible spongiform encephalopathy [9], autoimmune uveitis [10], acute graft-versus-host disease [11], inflammatory bowel disease (IBD) [12], and intestinal tumors. Therefore, it is of great importance to exhaustively survey the interaction between the physiological characteristics of DCs and diseases.

## 2. Types of DC

The DC family is very heterogeneous and consists of different DC subsets, each with specific functional characteristics. In general, two different DC subtypes can be distinguished, for example, myeloid DCs (mDCs) and plasmacytoid DCs (pDCs). These distinct subsets express various surface receptors and pattern recognition receptors (PRRs), which determine their specialized functions. The cDC subset can be identified and subdivided into three different subtypes by the expression of CD11c, in combination with their unique surface molecules CD1c (BDCA1), CD141 (BDCA3), and CD16. In addition, there are several special classifications, such as dendritic cells responding to specific microorganisms, CD14^+^ DC, microglia, moDCs, etc. Recently, another subset of DC, termed merocytic DCs (mcDCs), was defined and was found to likely resemble the cDC subset in mice [12]. It was shown that CD34^+^ hematopoietic stem cells produce granulocytes, monocytes, and progenitor cells of human DCs [13]. Rapamycin (mTOR) networks, which combine pattern recognition and growth factor receptor activation with nutritional information in cells and surrounding tissues, are critical in the proper development of mouse DCs [14]. In this section, we selectively focus on three subtypes of DCs, which may play a vital role in intestinal diseases.

### 2.1. cDCs

Conventional DCs, also called myeloid DCs (mDCs), express typical myeloid antigens CD11c, CD13, CD33, and CD11b in mice and CD11c in humans [15]. Mouse cDCs derive from common DC precursors in the bone marrow and comprise two main subsets, the CD8α^+^ and/or CD103^+^ cDC1 subset and the more heterogeneous CD11b^+^ cDC2 subset [16,17]. Human cDC1 expressed CD141/BDCA-1, CLEC9A, CADM1, and XCR1, and cDC2 expressed CD1c/BDCA-1, CD11c, CD11b, CD2, and FCER1 [18].

In mice, cDC1s expressed TLR2–TLR4, TLR11–TLR13, and C-type lectin-like receptor (CLEC)12A, and cDC2s expressed TLR1, TLR2, TLR4–TLR9, TLR13, retinoic acid-inducible gene I-like receptor (RLR), NOD-like receptor (NLR), STING, CLEC4A, CLEC6A, and CLEC7A [17]. In humans, cDC1s expressed TLR1, TLR3, TLR6, TLR8, TLR10, STING, and CLEC12A, and cDC2s expressed TLR1–TLR9, RLR, NLR, STING, CLEC4A, CLEC6A, CLEC7A, CLEC10A, and CLEC12A [17]. Tomer Granot [19] combined the technology of flow cytometry and fluorescence imaging, and indicated that human tissue CD13^hi^ CD141^hi^ Clec9A^+^ cells are cDC1s which maintain a stable quantity, whereas CD1c^+^Sirp-α^+^ cells are cDC2s with the specific ability to act as sentinel cells of mucosal tissues.

In the development of cDCs, basic leucine zipper ATF-like transcription factor 3 [20] and RELB proto-oncogene, which is also known as NF-κB subunit, are of great importance [21]. Mice lacking IFN-regulatory factor 8 (IRF8) [22], DNA-binding protein inhibitor ID2 [23], or nuclear factor IL-3-regulated protein (NFIL3) [24] exhibit a severe defect in the development of cDC1s. Some researchers found that although all these factors are required for CD103^+^ DC development, only IRF8 is essential for CD8α^+^ DCs [25]. Furthermore, cDC2 development is controlled by RELB63 and PU.1 [26].

cDC precursors travel through the blood to the lymphatic organs and peripheral tissues and then develop into immature DCs and migratory DCs, respectively [27]. These immature cDCs are dedicated to antigen sampling and are characterized by low levels of the expression of T cell co-stimulated molecules and MHCII classes, and then they may remain in resident tissues until an activation signal is encountered [28]. Recent research revealed that tissue-specific factors, including ATF-like transcription factor 3 and RELB proto-oncogene, program the expression levels of different proteases during DC differentiation, thus conferring tissue-specific functions to the different DC subsets in the spleen or thymus [29].

### 2.2. pDCs

In mice, pDC expressed Siglec-H, bone marrow stromal antigen-2 (BST2), lymphocyte activation marker 3 (LAG-3), B220, Ly6C, CD11c, CD8α, and Ly49Q [29]. Human pDCs are usually short of myeloid antigen, expressing CD45RA, CD4, and variable CD2 and CD7, and are distinguished by the expression of CD123, CD303, CD304, and ILT7 [15,30,31].

Research revealed that human pDCs are composed of two subsets, which can be distinguished by the expression of CD2. CD2^hi^ pDCs are involved in initiating T cell immune responses and can be detected in some tumor biopsies, whereas CD2^low^ pDCs display a limited capacity to induce allogeneic T cell proliferation [32]. pDCs are mainly developed from IL-7R^+^ lymphoid progenitor cells with the assistance of transcription factors E2-2 and IRF8 [19,33]. pDCs selectively express TLR2, TLR7, TLR9, TLR11, amd TLR12, and it is endosomal sensors TLR7 and TLR9 that mainly mediate the recognition of microbial pathogens [34].

Compared to cDCs, pDCs have a limited effect on antigen presentation and are always regarded as immunomodulating cells [35]. pDCs sense and respond to viral infection by rapidly generating numerous type I and III interferons and secreting cytokines [18,36]. Instead of leaving from bone marrow and differentiating in peripheral organs like cDC progenitor cells, pDCs differentiate completely in the bone marrow. The transcription factor Runx2, specifically expressed in an E2-2-dependent manner in the pDC, is necessary for the migration of the pDC from the bone marrow to the peripheral organs [37]. As considerable efforts have been focused on cDCs, some scientists have proposed that pDCs may be used in therapy to fight against cancer, because pDCs express a wide variety of PRRs, which can be harnessed to facilitate the targeted delivery of antigens to pDCs, leading to antigen presentation and activation of both CD4^+^ and CD8^+^ T cells [38].

### 2.3. moDCs

Human moDCs express CD13, CD33, CD11b, CD11c, CD172a, and MHCII36. Furthermore, short chain fat acids (SCFAs) have immunomodulatory effects on moDCs, which downregulate CXCL9, CXCL10, and CXCL11 production [39], and SCFA receptors such as GPR41, GPR43, and GPR109A are expressed on the surface of human monocyte-derived DCs.

During colitis in mice, a large amount of Ly6Chi mononuclear cells (MOs) can invade the colon and then differentiate into proinflammatory CD103^−^ CX3CR1^int^ CD11b^+^ DCs, producing IL-12, IL-23, iNOS, and TNF [18,40]. Some investigations demonstrated that moDCs (CD11c^+^ MHCII^hi^ cells that were also CD86^hi^ and F4/80^lo^) were specifically produced by monocyte-dendritic cell progenitors and not granulocyte-monocyte progenitor-derived Ly6C^hi^ MOs [41]. The process of producing moDCs requires granulocyte/Mφ colony stimulating factor (GM-CSF) [27]. Other studies have shown that constitutively migrating LY6C^+^ MOs can maintain their own properties instead of differentiating into DCs [42]. In this respect, a further study revealed that Ly6C^+^ cells include three sub-communities, among which CD11c^-^ Flt3^+^ MOs may be the precursor of CD209a^+^ PDL2^+^ moDCs, promoted by the increase in transcription factor PU.1 [43]. The summary of DC types refers to Figure 1 and Box 1.

Box 1Types of DC.DCs have two main types (mDCs and pDCs) and several special classifications, like DC responding to specific microorganisms, CD14^+^ DC, microglia, moDC.cDC1 and cDC2 are the subsets of cDCs, with different molecules and functions.Both cDCs and pDCs contain two subtypes that require a variety of immune factors and immune cells to assist their growth or function.

## 3. Physiological Function of DCs in the Gut

Approximately 1000 trillion microbes, composed of an estimated 4000 strains, are living in the human intestine and feeding on food residue from the digestive tract [44], producing complicated metabolites to regulate human physiological responses. Usually, DCs come into contact with intestinal flora through two different ways. The first model requires a special intestinal epithelial cell called the M cell. Commensal bacteria and other intact antigens discharged by M cells are captured by DCs in Peyer’s patches, which migrate to mesenteric lymph nodes (MLNs) and activate B cells in mesenteric lymph nodes, eventually resulting in IgA production [45,46,47]. Another model assumes that DCs directly sample symbiotic bacteria and/or other intact antigens in the lumen. The pioneering research supporting this model is the discovery that DCs extend dendrites into the lumen by penetrating the epithelium for tight connections [48].

DCs in the intestinal lamina propria (LP) of mice can be phenotypically divided into two major developmentally distinct populations, which are MHCII^+^ CD11c^hi^ CD103^+^ CD11b^+^ CX3CR1^−^ M-CSFR^lo^ DCs (CD103^+^ CD11b^+^ DCs) and MHCII^hi^ CD11c^hi^ CD103^−^ CD11b^+^ CX3CR1^+^ M-CSFR^hi^ DCs (CD103^−^ CD11b^+^ DCs) [49]. Previous studies have shown that the DC assembly transports autoantigen from the ileum to the T-cell region of the mesenteric lymph node to maintain the immune response [50]. In addition, DCs are of the utmost importance in peripheral immune tolerance through the finely regulated DC and Foxp3^+^ Treg cell crosstalk, whereby Treg cells modulate DC phenotype and function. CD103^+^ DCs drive the differentiation of Foxp3^+^Treg cells from CD4^+^ T cells depending on retinoic acid (RA) through RALDH2 and TGF-β, with the assistance of integrin αvβ8 [51,52,53]. RALDH2, an enzyme expressed by cDCs in the gut and encoded by *aldhla2*, can produce RA from dietary vitamin A [54]. In the absence of CD103^+^ DCs, RALDH2 expression and Treg production were decreased, but Th1 response was enhanced [55] and the mechanisms for this change remain unclear. In colitis, the efficiency of producing Treg by MLN DCs is lower because of the specific loss of CD103^+^ DCs [56]. CTLA-4 expression in Treg cells down-/upregulates CD80/86 co-stimulatory molecules, which are significant for the activation of the immune response on DCs [51].

In some bacteria like *L. reuteri* and *L. casei*, IL-10 is also involved in inducing tolerogenic setting-predominated DC subtypes [53]. IL-10 also can inhibit multiple aspects of DC function including MHCI/II and CD80/CD86 co-stimulatory molecule expression and the release of pro-inflammatory cytokines [57]. In addition, IL-10 production by human DCs is triggered by Treg cells, which stimulate B7-H4 expression and render APCs immunosuppressive [58]. The immunomodulatory molecule, polysaccharide A (PSA), of *B. fragilis* mediates the conversion of CD4^+^ T cells into Foxp3^+^ Treg cells that produce IL-10 during commensal colonization and suppress the activation and proliferation of inflammatory effector T cells [59]. Endogenous biomolecules like adrenomedullin, hepatocyte growth factor, immunoglobulin-like transcript, NF-κB, placental growth factor, TGF, TNF, VEGF, and several possible molecular mechanisms or signal pathways exert a considerable tolerogenic influence on DC function [60,61].

Tolerogenic DCs (tol-DCs), which consist of naive immature DCs or alternatively activated semimature DCs induced by apoptotic cells or the regulatory cytokine milieu, play a pivotal role in immune tolerance [62]. Tol-DCs constitutively migrate throughout the periphery and the lymphatic system, presenting self-antigens in the absence of costimulatory molecules [63]. Meanwhile, DC plays a certain role in the immune tolerance of the human body to intestinal microorganisms, which is related to programmed death receptor 1 (PD-1). PD-1 is a member of the B7 family, and human or mouse PD-1 ligand (PD-L) 1 and PD-L2 are expressed on immature DCs, mature DCs, interferon (IFN)-treated monocytes, and follicular dendritic cells [64]. Binding of PD-L1 to PD-1 leads to inhibition of T cell receptor (TCR)-mediated lymphocyte proliferation and cytokine secretion [65]. Moreover, mice with PD-L1^−/−^ developed autoimmune diseases, which indicated that peripheral tolerance was defective [66].

It is worth mentioning that unique tolerogenic properties are not only shaped by tissue-derived migratory CD103^+^ DCs, but also by resident lymph node (LN) stromal cells (SCs) [67]. A study has shown that mLN SCs are imprinted with a high Treg-inducing capacity soon after birth, and instruct LN-resident DCs (resDCs) to foster efficient Foxp3^+^ Treg induction in a Bmp2-dependent manner [68]. Bone morphogenetic protein (Bmp), a member of the TGF-β superfamily, has a synergistic effect with TGF-β on the induction of Foxp3^+^ iTreg [69]. These regulatory molecules or cells mentioned above contribute to the immunity tolerance caused by DCs.

## 4. Regulatory Relationship between the Gut and DCs

In most tissues, exposure to microbial products is sufficient to convert immature cDCs into mature cDCs, thereby producing an effective effector response. However, it is likely to be common that symbiotic bacteria expose their PAMPs in the healthy intestine. How the intestine can tolerate trillions of intestinal bacteria, initiate tolerance toward food antigens, and fight infections is the subject of an intense area of research. Recent advances have highlighted a fundamental role of mouse DCs in these functions.

Numerous studies have shown that exposure to PAMPs present on intestinal commensal bacteria promote DCs to express a unique molecular footprint so as to promote the differentiation of naive B2 cells into IgA, producing plasma cells with the help of RA and TGF-β [70,71]. IgA secreted by plasma cells effectively limits the penetration of commensal intestinal bacteria and opportunistic pathogens. Other studies have provided further evidence that stimulation of early bacterially exposed cells results in increased IL-10 secretion and the inhibition of DC differentiation through the MyD88 signaling pathway, leading to functional suppression [72].

Apart from the influence of intestinal flora, epithelial cells can also be affected by the condition of mucosal dendritic cells through the constitutive release of thymic stromal lymphopoietin (TSLP) and TGF-β. Commensal bacteria via microbe-associated molecular patterns (MAMPs) bind to TLRs on intestinal epithelium cells (IECs) and DCs, and upon activation of TLR signaling, IECs release TSLP and TGF-β [73]. TSLP and TGF-β cooperate to elicit the tolerogenic phenotype of DCs, as well as promoting the polarization of T cells toward a noninflammatory Th2 response [74,75]. Mincle, a Syk-kinase-coupled C-type lectin receptor, and Syk signaling couples the sensing of mucosa-associated bacteria by DCs in PPs with the production of IL-6 and IL-23, cytokines that regulate IL-22 and IL-17 production through T cells and innate lymphoid cells, thus promoting the intestinal immune barrier and limiting microbial translocation [76]. A common downstream signaling adaptor, caspase recruitment domain 9 (CARD9), is reported to regulate the gut mycrobial and bacterial landscape [77,78]. Based on the above, the explain for the general mechanism of DC immune tolerance is clear, which is of great importance in commensal intestinal bacteria colonization.

However, a study has demonstrated that the absence of CD103^+^ CD11b^+^ LP DCs, as well as reduced numbers of Th17 and Treg cells in the LP, does not significantly affect the composition of the steady-state enteric microflora [79]. This may demonstrate the importance of parental transmission and early bacterial explosion we mentioned before, which also deserves further study.

Germ-free (GF) mice showed delayed development of intestinal DC, indicating the important role of intestinal microbes in intestinal immunity [80,81]. Some researchers report that moDCs are able to mediate the responses of robust T helper cells (Th) 1 and Th17 upon stimulation by *Escherichia coli Schaedler* or *Morganella morganii*, whereas the probiotic *Bacillus subtilis* strain limits this effect [82]. Some studies indicate that lipid-regulated nuclear receptor peroxisome proliferator-activated receptor γ (PPARγ) plays a major role as a positive transcriptional regulator in human developmental DCs, mainly by controlling the genes involved in lipid metabolism and thereby indirectly modifying the immune phenotype [83]. Other researchers attached the importance of vitamin A to intestinal homeostasis, because cytokine-activated colonic epithelial cells trigger the secretion of distinct combinations of chemokines depending on the pro-inflammatory stimulus and are controlled by RA [84]. In addition, the outcome of moDC differentiation is able to accommodate to unique cellular microenvironments and remains remarkably plastic until the terminal differentiation of the moDCs ensues [82]. *Bifidobacteria* can significantly improve the antigen uptake and processing capacity of DCs in Crohn’s disease (CD) patients, which can partially solve the reduction of intestinal innate immunity and reduce the uncontrolled microbial growth in the intestinal tract of children with IBD [85]. Moreover, *bifidobacteria* are able to directly cause the production of DC maturation and cytokines, and activate the production and maturation of DCs [86], which in turn supports improved effector function of tumor-specific CD8^+^ T cells [87]. Researchers studied a symbiotic microorganism called *Bifidobacterium longum subspecies infantis* 35,624 and found that it can be recognized by mDCs and pDCs through different PRRs, inducing Foxp3T regulatory cells through different ways of maintaining intestinal homeostasis [88].

Intestinal bacteria have attracted extensive attention in the field of digestion, and it is worth mentioning that the role of enterovirus in intestinal homeostasis has gradually been recognized and verified by researchers. Researchers suggested that pretreatment with antiviral cocktail therapy can lead to severe colitis, and found that resident intestinal viruses played a protective role in gut inflammation through TLR3 and TLR7-mediated IFN-β secretion by pDCs [89]. On the other hand, several studies have shown that TLR3 agonist poly I:C pretreatment can improve the therapeutic effect of umbilical cord mesenchymal stem cells in dextran sulphate sodium (DSS)-induced colitis [90,91,92]. Importantly, there were disease-specific changes in the enterovirus in IBD, which is a significant expansion of the taxonomic richness of *Caudovirales* bacteriophages, and the viruses appear to be different, although the changes have been observed in both Crohn’s disease (CD) and ulcerative colitis (UC) [92]. In addition, during intestinal tumor therapy, DCs from the combination treatment group of radiotherapy and vancomycin significantly increased when compared with radiotherapy, which indicates the potential relationship between intestinal bacterial and DCs [93]. The summary of the regulatory relationship between gut and DC refers to Figure 2 and Box 2.

Box 2Crosstalk and interaction between the gut and DCs.DCs come into contact with intestinal flora within or without the assistance of microfold cells.CD103^+^DCs help the establishment of immune tolerance through the effect of T cells.IL-10, PD-1, and many other factors play roles in immune tolerance.Differentiation and proliferation of DCs can be influenced by intestinal flora, epithelial cells, and viruses.

## 5. Intestinal DCs and IBD and Intestinal Tumors

### IBD and DCs

IBD, including UC and CD, is a chronic inflammatory disease caused by microbial invasion or mucosal barrier damage, observed primarily in genetically susceptible populations. In recent years, tremendous investigation has proven the development of the pro-inflammatory effect of DC cells of mice on IBD, considering DCs as a potential target for the treatment of IBD [11,94].

A recent study suggested that the maladjustment of the Th1 response in the inflammatory colonic mucosa of IBD patients was caused by a change in PD-L1 expression in the mucosal mesenchymal stromal cell compartment, because increased PD-L1 expression inhibits Th1 cell activity in UC, whereas the loss of PD-L1 expression observed in CD leads to the persist existence of a Th1 inflammatory milieu [95]. Although the experiment focused on CD90^+^ stromal cells, the expression of PD-L1 in blood DCs did increase during CD. The detailed mechanism needs to be further studied. Similar studies have demonstrated the protective effect of PD-L1 mediated inhibitory signals during colon inflammation [96]. PD-L2, another ligand of PD-1, is expressed on DCs and through PD-L1 indirectly promotes the secretion of the proinflammatory cytokines TNF-α and IFN-γ, which are known to cause the pathogenesis of the disease and are involved in the progression of CD [97].

Furthermore, the role of pDCs in IBD has attracted the attention of scientists. Some researchers have demonstrated that the circulating pDCs in active IBD patients migrate to secondary lymphatic organs and inflammatory sites, secreting inflammatory cytokines such as IL-6, IL-8, and TNF-α [98,99]. It may lead to a pro-inflammatory phenotype in T cells (Th1), which helps to understand why the inflammation is perpetuated in IBD [100].

Experiments on mice showed that at homeostasis, pDCs can migrate to specific tissues, such as the lymph gland or gut, under the control of chemokine receptors including CCR5, CCR7 [101], CCR9 [102], and its corresponding ligands CCL19 (MIP-3β), CCL21, integrin α4β7, and CCL25 [103]. During the development of DSS-induced acute colitis in mice, the transcriptional expression level of CCL2, CCL3, CCL5, CCL7, CCL8, CCL25, CXCL9, CXCL10, and CXCL11 is increased [104]. These results suggest that pDCs may be chemokine-dependent and transported in the inflammatory laminae of the colon. However, through targeting the pDC-specific transcription factor TCF4 (E2-2) in experimental IBD caused by deficiency of Wiskott–Aldrich syndrome protein (WASP) or IL-10, some researchers proposed that pDCs do not play a major role in the pathogenesis of intestinal inflammation in IBD [105]. The differences regarding the roles of pDCs may be due to the use of pDC ablation systems and a genetic IBD model, which is T-cell-dependent, a different colitis model from T-cell-independent colitis induced by DSS [106]. Although the role of pDC in IBD may be controversial and deeper and more accurate studies are needed, the potential therapeutic value of pDCs in IBD cannot be completely ignored.

cDCs, another subset of DCs, may also play a role in the development of IBD. CD11c^+^ DCs, a subtype of cDCs mentioned above, were reduced by over 75% in the inflamed and uninflamed ilea in patients with CD compared to controls, as measured through multicolor tyramide fluorescent labeling with automated analysis, and these non-inflammatory areas showed no visible damage or inflammation, suggesting that the loss of DCs may be a precursor to subsequent damage [97]. In terms of health, the phenotypic DC subsets found in the intestinal mucosa maintain their tolerance and switch to a proinflammatory phenotype during infection or chronic IBD [107]. Dab2, a clathrin and cargo binding endocytic adaptor protein, modulates several cellular signaling pathways, including TGF-β [108], and has been linked to acute or chronic inflammation [109,110]. It was also found that Dab2 is mainly expressed in intestinal CD11b^+^ DCs, and the ablated expression of Dab2 in DC2.4 cells (murine immortalized dendritic cells)—measured using the CRISPR-CAS9 system—intensifies exacerbated experimental colitis [107]. In the presence of IL-15, all-trans RA induces the release of pro-inflammatory IL-12 and IL-23 by cDCs and promotes intestinal inflammation [111].

In contrast, some researchers found that CD103^+^ cDCs in patients with UC had acquired a potent ability to drive Th1/Th2/Th17 cell responses, which are associated with increased expression of pro-inflammatory cytokines [112]. The frequency of CD103^+^ cells among cDC1 and cDC2 subsets was lower in active IBD intestinal tissue compared to controls [113]. In mice, experiments show that the expression of p38α in CD103^+^ DCs regulated the balance between iTreg and Th1 differentiation in a TGF-β2-dependent manner [114]. A recent study revealed the further mechanism by which CD137, a potent costimulatory receptor for CD8^+^ T cells, can participate in activating TAK1 and subsequently stimulates the AMPK-PGC-1α axis to enhance expression of Aldh1a2, which encodes RALDH2 [115]. Furthermore, some studies revealed intestinal CD103^+^ CD11b^-^ DCs could inhibit colitis through epithelial anti-inflammatory response induced by IFN-γ [116,117]. In addition, research shows that the frequency of αVβ8^+^ cDC2s, which are thought to induce Tregs, increased in Crohn’s patients compared to controls [118]. The evidence above shows the role of cDCs in IBD.

It can thus be seen that intestinal DCs may become a new treatment for patients with IBD and therefore warrant further study. For example, the latest studies proved that by transfecting primary rat bone marrow DCs with FasL plasmid into the peritoneum of IBD model rats, intestinal damage is reduced and the number of colon T cells, neutrophils, and pro-inflammatory Mφs decreased [119]. The summary of relationship between DC and IBD refers to Figure 3. 

The relationship between DCs and tumors has been extensively studied, and the use of a “DC vaccine” has even been proposed to stimulate the entry of effector T cells into the tumor to suppress the tumor [120,121,122]. DC subsets are also considered to be predictive indictors of gastric cancer prognosis [123]. Some researchers suggested that a conserved DC regulatory program, which was related to the capture of cell-associated antigens in normal or excessive cell death, inhibited immune function and controlled the threshold of T-cell activation [124]. Inflammatory mediators and effector cells are important components of the tumor local microenvironment. In some types of cancer, chronic inflammation precedes malignant changes, and in others, carcinogenic changes trigger inflammatory microenvironments that promote tumor development [125,126].

Impaired numbers and functions of DCs have been widely observed in several types of cancer, including colon cancer, which may be related to tumor escape mechanisms that cancer cells use to evade host immune surveillance [127,128,129]. Tremendous studies have shown that the tumor microenvironment can act on human pDCs through immunosuppressive mediators (such as PGE2 and TGF-β [130]) or pDC regulatory receptors to inhibit or alter its functional activity, possibly leading to inhibition of IFN-α secretion or induction of Treg and preventing an effective anti-tumor response [131,132,133]. The IL-10 derived from tumors can also repress the antitumor immunity of human DCs [134]. In a murine breast cancer model, IL-10R was expressed at high levels on DCs, leading to the suppression of the anti-tumor cytokine IL-12 [121]. Another discovery was that the dysfunction of DCs may be related to lipid accumulation in human and mouse DCs caused by the up-regulation of scavenger receptor A, and thus DCs fail to effectively stimulate allogenic T cells or the presence of tumor-related antigens [135,136].

Another subset of DCs, inflammatory DCs (inf-DCs), is a subtype of moDCs described earlier, and can effectively capture and deliver tumor antigens [137,138]. Human inf-DCs are known to induce Th17 cells in inflammation and are one of the major sources of IL-17 [139]. IL-17R family members in inflammatory environments may play a role in the tumor-promoting effect of IL-23 [140]. Moreover, the barrier damage caused by the tumor triggers the inflammation caused by the tumor and promotes the growth of the tumor [141].

Inf-DCs can also produce reactive oxygen species (ROS) and directly mediate the anti-tumor response in mice [142]. Experiments on mice show that specific dynamic fluctuations in the expression of AT-rich sequence-binding protein 1 (Satb1) control the generation and immune stimulation activity of homeostasis DCs and inf-DCs, and the continuous overexpression of Satb1 transforms them into carcinogenic/pro-inflammatory cells, thus promoting malignant progression [143].

The binding of co-stimulatory molecules CD80/CD86 on DCs induces certain subsets of DCs to express functional indoleamine 2,3-dioxygenase (IDO), an enzyme that degrades essential amino acid tryptophan (Trp) into kynurenine (Kyn) [144]. IDO1-mediated tryptophan catabolism promotes local immune suppression in two ways [145]. The first is that tryptophan starvation limits T cell proliferation by weakening the T cell cycle mechanism, whereas the other one depends on the apoptosis of T cells caused by Kyn [146]. Wnt signaling networks in DCs that drive Treg responses have also been proven to be involved through several complicated signal axes, including molecules like β-catenin, RA, mTOR, IDO, and IL-10 [147].

In addition to the effect of DCs on tumors, γδT cells also have shown the ability to promote tumor progression by interfering with DC effector function, which can be mediated by bacteria through IL-1β and IL-23 [148]. As for the further mechanism, it is speculated that γδT cells suppress innate and adaptive immunity through the induction of immunosenescence, which significantly upregulated PD-L1 expression and resulted in other impaired phenotypic and functional features [134,149,150].

Another molecule mentioned above, PD-1, may play a vital role in tumor progression, as the antibody of PD-L1 can enhance the maturity of DCs and subvert the immunosuppressive state of the tumor microenvironment [151]. Anti-PD-1 cancer immunotherapy is based on the crosstalk between T cells and DCs, and the progress is licensed by IFN-γ and IL-12 [152].The summary of DC’s contact with intestinal tumor refers to Figure 4 and the contact with intestinal diseases refers to Box 3.

Box 3Diseases and DCs.The detailed role of pDCs in IBD is controversial because of different experimental models.cDCs are involved in the occurrence and development of IBD through several molecular processes like RA, IL-12, and IL-23 and influence the differentiation of T cells.inf-DCs and PD-1 are vital to intestinal tumors through multiple immune factors.

## 6. Conclusions

The evidence above sufficiently demonstrates the importance of intestinal DCs, which may have huge potential for being applied in clinical treatments. DCs have several subgroups in different classification cases, including cDCs, pDCs, moDCs, and others, all of which participate in different regulatory networks. The regulatory relationship of DCs and intestinal microbiota provides a possible and easier way to access and influence DCs in the gut or even around the body. DCs are also involved in the immune tolerance to symbiotic bacterial colonization through Foxp3^+^ Treg cells, which require TGF-β, IL-10, RA, and many other biomolecules. Additionally, multiple molecules are combined in the complicated mechanism of regulating intestinal diseases such as IBD and intestinal tumors by DCs, which also makes it more difficult to achieve satisfactory therapeutic effects merely by altering several molecular targets. Collectively, DCs have shown large potential and warrant further and deeper studies from the prospective of intestinal physiology.

## Figures and Tables

**Figure 1 cells-10-00583-f001:**
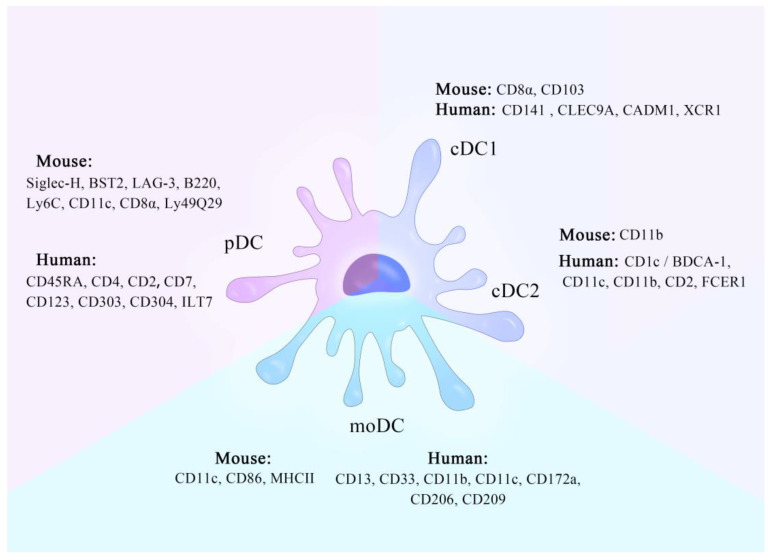
A schematic diagram of characters of different dendritic cell (DC) subsets. There are three DC subtypes, which are conventional DCs (cDCs), plasmacytoid DCs (pDCs), and DCs derived from monocytes (moDCs). cDCs can be divided into two subtypes, cDC1 and cDC2. They can be distinguished depending on the specific molecules on the human or mouse DC surface.

**Figure 2 cells-10-00583-f002:**
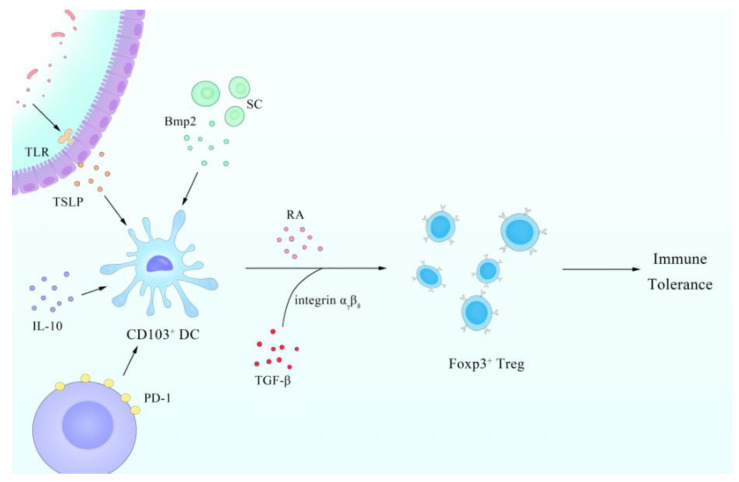
Summary of the role of CD103^+^ DCs in intestinal immune tolerance. The immune tolerance in the intestine is mainly due to the Foxp3^+^ Treg. CD103^+^ DCs play a significant role in regulating Foxp3^+^ Treg, in which TGF-β and retinoic acid (RA) are important signaling molecules. IL-10, PD-1, and many others are also involved and have effects on DCs. In addition, on intestinal epithelium cells (IECs) and stromal cells (SCs) can promote the effect of CD103^+^ DCs after activation.

**Figure 3 cells-10-00583-f003:**
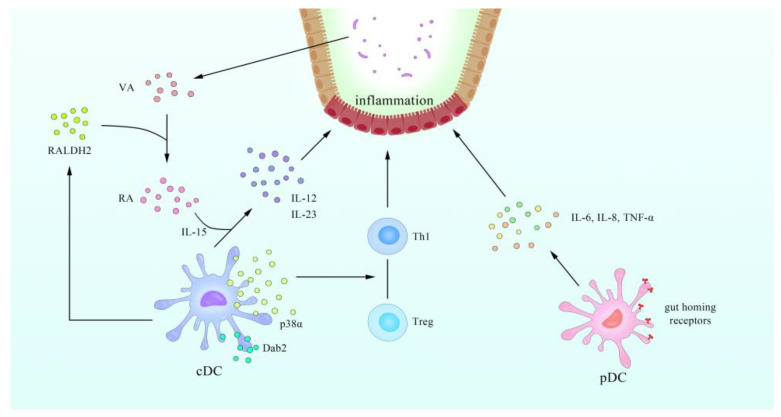
A schematic overview of the regulatory relationship between DCs and inflammatory bowel disease (IBD). The lost integrity of the epithelial cell barrier and insufficient mucus layer facilitate bacterial translocation to subepithelial regions. Dab2 is expressed by cDCs. The expression of p38α in cDCs regulated the balance between iTreg and Th1 differentiation in a TGF-β2-dependent manner. RA, transforming from dietary vitamin A by RALDH2, induces the release of pro-inflammatory IL-12 and IL-23 by cDCs and promotes intestinal inflammation in the presence of IL-15. pDCs express gut homing receptors, which reside beneath the epithelial layer and are responsible for taking up luminal antigens and gaining the capacity to produce IL-6 and IL-8 TNF-α, which plays a key role in the establishment of IBD in model rats.5.2. Intestinal Tumors and DCs.

**Figure 4 cells-10-00583-f004:**
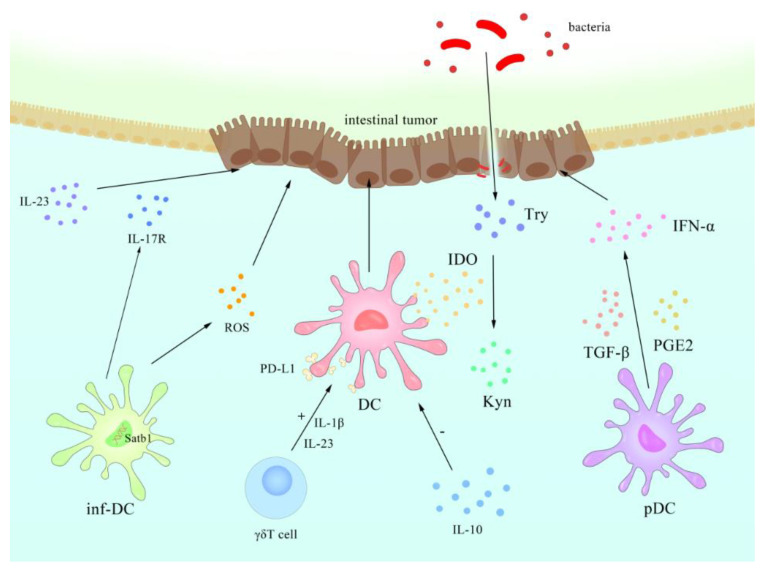
Schematic representation of known regulatory mechanism between DCs and intestinal tumors. The ensuing intestinal flora are disturbed by the frequent use of antibiotics and/or diet, stimulating inflammation that is largely orchestrated by DCs. Their activation by products of pathogenic bacteria induces Try, which in turn causes changes in Kyn. In the tumor microenvironment, immunosuppressive mediators such as PGE2 and TGF-β can act on pDCs to inhibit or alter their functional activity, leading to inhibition of IFN-α secretion. The IL-10 derived from tumors can also repress the antitumor immunity of DCs. γδT cells can suppress innate and adaptive immunity by interfering with DC effector function, which can be mediated through IL-1β and IL-23, and significantly upregulate PD-L1 expression. The continuous overexpression of Satb1 transforms homeostasis DCs and inf-DCs into carcinogenic/pro-inflammatory cells. Inf-DCs can also produce DNA-damaging ROS and directly mediate the anti-tumor response.

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
