# Peer review of "Functions of Dendritic Cells and Its Association with Intestinal Diseases"

_cells, 2021, doi:10.3390/cells10030583_

Round 1

Reviewer 1 Report

I would like to thank the authors for addressing my concerns and have no further comments.

Thank you

Author Response

Dear Reviewer,

Thank  you very much for your confirmation of our response  to the kind comments, which give us great assistence in the process of modified manuscript.

Best wishes

Jingdong Xu

Reviewer 2 Report

Concerning the answer to my comment:

 "Line 47: “Many researches demonstrated the potential link between DC and several common diseases.”. The authors shoud include a reference. Please see the following review for illustration: Coutant F et al, Altered dendritic cell functions in autoimmune diseases: distinct and overlapping profiles. Nat Rev Rheumatol 2016 Dec;12(12):703-715"

I do not understand the author's approach. I suggested to introduce, at this stage of the manuscript, a recent review illustrating the content of the sentence, not an endless list of original articles, as it has been done in the revised manuscript.   I think there is a confusion for this comment, since in the rebutal letter, the author writes: "Thanks for your generous advice.According to your advice, we have added the reference.Please see page 2 line 44"   ....However, it is not the case in the reviewed manuscript.

Author Response

1. "Line 47: “Many researches demonstrated the potential link between DC and several common diseases.”. The authors shoud include a reference. Please see the following review for illustration: Coutant F et al, Altered dendritic cell functions in autoimmune diseases: distinct and overlapping profiles. Nat Rev Rheumatol 2016 Dec;12(12):703-715"

Response: Thanks for your generous advice. According to your advice, we have added the reference and give more information for illustration in the manuscript. Please see page 2 line 43-46.

2. I do not understand the author's approach. I suggested to introduce, at this stage of the manuscript, a recent review illustrating the content of the sentence, not an endless list of original articles, as it has been done in the revised manuscript. I think there is a confusion for this comment, since in the rebutal letter, the author writes: "Thanks for your generous advice. According to your advice, we have added the reference. Please see page 2 line 44". However, it is not the case in the reviewed manuscript.

Response: We are very sorry for the confusion caused to you due to our unclear statement. According to your suggestion, we have revised the first sentence to express more clearly and simplify the following examples. We also add some information from the reference you mentioned. Thanks for your kind suggestions. Please see page 2 line 43-51 in bright blue.

This manuscript is a resubmission of an earlier submission. The following is a list of the peer review reports and author responses from that submission.

Round 1

Reviewer 1 Report

In this manuscript, Hua et al reviewed recent advances on the functions of DCs in intestinal diseases. The review is interesting, but it should be presented in a clearer manner, by explaining in each section what is known in humans and what is known in mice. For instance, in the first section entitled “types of DC”, the authors should detail the phenotype and the functions of each DC subsets both in humans and in mice. Then, for each section of the review, the authors should specify what has been demonstrated in humans and in mice.

There following are some concerns:

  • Line 47: “Many researches demonstrated the potential link between DC and several common diseases.”. The authors shoud include a reference. Please see the following review for illustration: Coutant F et al, Altered dendritic cell functions in autoimmune diseases: distinct and overlapping profiles. Nat Rev Rheumatol 2016 Dec;12(12):703-715.
  • Line 65 MoDC: please define

Reviewer 2 Report

The manuscript presented by Yang et al is a review aimed to explain the “Functions of dendritic cells and its association with intestinal diseases”. While some passages are interesting, such as section 2, 3 and 4.1, the review is generally very superficial and sometimes simplifying too much for our current scientific knowledge. Foremost, the complexity of the dendritic cell family is hardly explained and it is unclear when the authors refer to human or mouse DCs. Moreover, in every section, the authors focus on selected examples of the current literature, but do not provide a comprehensive overview or summary. This should at least be mentioned, best in the abstract. Additionally, the English writing and grammar has to be largely revised. Currently, the manuscript is quite challenging to read and many sentences are unfortunately very difficult to understand and give confusing information. Currently, the presented manuscript requires a major revision before potential publication in Cells. Please see my more detailed additional comments below.

Highlight section

What are “tissue-derived migratory CD103+DCs”?

  1. Introduction section

Please define all abbreviation when they are first mentioned, for example “Mφs”, “PPRs”, “SCFA”, “CD”, “UC”, “IBD” and many more. Also, MHCII is the common abbreviation used for major histocompatibility complex II, not MHCII.

Line 31: what are “specific immune cells”? Do the authors mean adaptive?

Line 39: please indicate that there is a large group of pattern recognition receptors containing different classes of receptors. TLRs are only one of them!

The following sentence completely lack context “At the same time, researches have shown that the circulating DCs of celiac disease patients instead of pDC exhibited intestinal homing characteristics and were regulated by different gliadin-derived peptides” What do the authors mean, how does this information fit with the resot of the paragraph?

Please proved more references for the following statement “During embryonic development and postnatally, DCs progenitors migrate into non-lymphoid organs and differentiate into immature DCs, which gradually form a dense network of sentinel cells at all outer and inner surfaces of the body, as well as in most organs.”. This process is not well understood at all and this sentence represents an overstatement.

Paragraph Line 47-55: Please state that the mentioned pathologies are selected examples for the importance of dendritic cells in disease and, please, make sure they are properly referenced. For example, there is no reference provided for “tumor” in line 53.

  1. Types of DC section

Please give more details on the DC types, especially if the authors refer to mouse or human subsets. It is very important DCs are commonly classified according to their ontology (PMID: 25033907, 31467405, 24837101) of being derived from the common dendritic cell precursor in the bone marrow or monocytes, which the authors not even mention. Neither do they mention the cytokines or transcription factors involved – which should crucially be added (see references above). Following this rationale, Langerhans cells rather represent Macrophages than classical dendritic cells (PMID: 26554892).

  • cDCs section

Again, it is not clear whether the authors refer to human or mouse DCs. Some information is simply not really correct, such as the surface marker expression and the explanation of cDC1s and cDC2s (see references PMID: 25033907, 31467405, 24837101). Please extend the information given on cDC1s and cDC2s.

Why are the lineage transcription factors for cDC1s and cDC2s not mentioned, if those are later discussed for pDCs? Same for PRRs? The authors should refer to the comprehensive table in PMID: 31467405.

Also, why is Reference 20 cited here? This does not appear to be the correct reference for this sentence. Please revise.

  • pDCs section

Please indicate if mouse or human DCs are discussed! Unfortunately, there are again several misconceptions contained in this section of the manuscript. It is important to mention that the antigen-presenting role of pDCs is very limited compared to cDCs (PMID: 31467405, 23516985, 24837101). The manuscript gives the opposite impression.

Moreover, the following sentence is unfortunately very incomplete, if not again incorrect information “Some scientists reckon that pDCs can be used as therapy to fight against cancer, for pDCs express a wide variety of pattern recognition receptors (PRR) which can be harnessed to facilitate the targeted delivery of antigen to pDCs, leading to antigen presentation and activation of both CD4+ and CD8+T cells.”. While pDCs are pursued in cancer therapy, the main effort of the scientific community focussed on cDCs due to their superior antigen-presentation potential (.PMID: 31467405, 23516985, 24782868, 28192720, 22566899, 29119057).

  • moDCs section

Please indicate if mouse or human DCs are discussed! Again, there are several examples given and it should be indicated that this are only some examples and not a complete overview of the literature.

Figure 1: please add the information of “subtypes” to moDCs and the information of “Secreting” to cDCs. Please generally separate cDCs into the 2 widely recognized subsets cDC1 and cDC2 (PMID: 25033907, 31467405, 24837101).

Box 1: cDC1s and cDC2s have to be distinguished as they are recognized for a very long time and very different cell types (PMID: 25033907, 31467405, 24837101).

  1. Physiological function of DCs in gut section

Please indicate if mouse or human DCs are discussed! Again, there are several examples given and it should be indicated that this are only some examples and not a complete overview of the literature. See, for example, a more comprehensive overview in PMID: 27760337 (Type 3 Immune responses), although more recent references should be added.

Bmp is “Bone morphogenic protein”. Please indicate which Bmp.

Please spell “Peyer's patches” correctly

  1. Regulatory relationship between gut and DCs section

Please indicate if mouse or human DCs are discussed! Again, there are several examples given and it should be indicated that this are only some examples and not a complete overview of the literature. It is a pity that the authors only took TLRs into consideration. The importance of other PRRs for the functions of intestinal dendritic cells should definitely be discussed. For example, the role of c-type lectins such as Mincle (PMID: 30709742) and other DC expressed receptors should definitely be included in the review.

  1. Intestinal DCs and IBD & intestinal tumor section

Please indicate if mouse or human DCs are discussed! Again, there are several examples given and it should be indicated that this are only some examples and not a complete overview of the literature.

Please also discuss cDC1s and cDC2s separately!

Importantly, the following sentence appears conceptually wrong and should be rephrased “PD-L2, another ligand of PD-1, is expressed on DC and distinctly promotes the secretion of the proinflammatory cytokines TNF-α and IFN-γ, which are known to cause the patho-genesis of the disease and involve in the progression of CD.” It is not that PD-L2 is promoting TNF-α and IFN-γ production in the intestine as the authors claim, PD-L2 is also largely an immune-dampening or modulating surface molecule (PMID: 22611421). What the paper Faleiro et al. actually suggests is that enhanced PD-L1 expression on DCs usually dampens TNF-α and IFN-γ expression. However, PD-L2 (which is also upregulated) can “compete” with PD-L1 for binding of PD-1 and thereby indirectly “prevents” inhibition of TNF-α and IFN-γ. However, by no means, I understand from the cited study that PD-L2 is actually promoting the expression of those cytokines. Please correct the sentence in the main text accordingly

Please revise this sentence, it is very difficult to understand “CD11c+DC, subtype of cDC mentioned above, induced a reduction of 75% in the inflamed and uninflamed ileum in patients with CD compared to controls through multicolour tyramide fluorescent labelling with automated analysis, and these non-inflammatory areas showed no visible damage or inflammation, suggesting that the loss of DC may be a precursor to subsequent damage.”

How does the deletion of a gene in vitro cause colitis??? As the authors wrote “It was also found that Dab2 is mainly expressed in intestinal CD11b+DC, and the ablated expression of Dab2 in DC2.4 cells (murine immortalized dendritic cells) through using the CRISPR-CAS9 system resulted in exacerbated experimental colitis.

Also, please give more details on the section “The latest researches proved that by transfecting primary rat bone marrow DC with FasL plasmid into the peritoneum of IBD model rats, it reduced intestinal damage and the number of colon T cells, neutrophils and pro-inflammatory Mφs decreased”, currently it is written too confusing to understand.

Lastly, in the section about cancer, it is important to specify which mechanisms have been reported in intestinal cancer, currently the section is a very selective and incomplete summary of some pieces of literature. Please see PMID: 31467405.